# Fournier’s Gangrene as an Adverse Event Following Treatment with Sodium Glucose Cotransporter 2 Inhibitors

**DOI:** 10.3390/medicina60050837

**Published:** 2024-05-20

**Authors:** Ioana-Maria Suciu, Alin Greluș, Alina-Ramona Cozlac, Bogdan-Simion Suciu, Svetlana Stoica, Silvia Luca, Constantin-Tudor Luca, Dan-Ion Gaiță

**Affiliations:** 1Institute of Cardiovascular Diseases Timișoara, 13A Gheorghe Adam Street, 300310 Timișoara, Romania; 2Institute of Life Sciences, “Vasile Goldiș” Western University of Arad, Str. Liviu Rebreanu 86, 310045 Arad, Romania; 3Arad County Emergency Clinical Hospital, Str. Andreny Karoly nr. 2–4, 310037 Arad, Romania; 4Cardiology Department, “Victor Babeș” University of Medicine and Pharmacy, 2 Eftimie Murgu Sq., 300041 Timișoara, Romania; 5Research Center of the Institute of Cardiovascular Diseases Timișoara—IBCV-TIM, 13A Gheorghe Adam Street, 300310 Timișoara, Romania

**Keywords:** sodium glucose cotransporter 2 inhibitors, Fournier’s gangrene, necrotizing fasciitis, serious adverse events, diabetes mellitus

## Abstract

We present the case of a 51-year-old male with known congestive heart failure and acute myocarditis who presented to the emergency department (ED) with swollen testicles and urinary symptoms two weeks after the initiation of sodium glucose cotransporter 2 (SGLT2) inhibitor treatment. Abdominal and pelvic computed tomography (CT) scan was consistent with the diagnosis of Fournier’s gangrene (FG). Intravenous antibiotics were administered and surgical exploratory intervention and excision of necrotic tissue were performed, stopping the evolution of necrotizing fasciitis. FG, a reported adverse event, may rarely occur when SGLT2 inhibitors are administered in patients with diabetes. To our knowledge, there have been no reported cases of FG in Romania since SLGT2 inhibitors were approved. The distinguishing feature of this case is that the patient was not diabetic, which emphasizes that patients without diabetes who are treated for heart failure with SGLT2 inhibitors may also be at risk of developing genitourinary infections. The association of predisposing factors may have contributed to the development of FG in this case and even though the benefits of SGLT2 inhibitors outweigh the risks, serious adverse events need to be voluntarily reported in order to intervene promptly, verify the relationship, and minimize the risk of bias.

## 1. Introduction

Sodium glucose cotransporter 2 (SGLT2) inhibitors were developed to be an additional class of anti-diabetic drugs but are now used alongside other novel glucose-lowering agents to treat a whole spectrum of heart failure conditions regardless of the presence of type 2 diabetes mellitus (T2DM). By inhibiting SGLT2, the renal threshold for glucose decreases, interfering with tubular glucose reabsorption [1,2]. Glucose will promote the excretion of free water via osmotic diuresis and the excess glucose remaining in the tubule will be excreted in urine, leading to glycosuria (70–80 g/day), which predisposes the patient to bacterial infection [3]. Although no additional complication is desired, a list of serious adverse events limits the prescription of SGLT2 inhibitors in some patients [2,3]. Genitourinary infections are reported adverse events that can occur after the administration of SGLT2 inhibitors. Serious urinary tract infections (UTI), such as urosepsis and pyelonephritis, as well as various genital infections, such as genital warts, mycotic infections, urethritis, or balanitis, have been reported. Despite the fact that SGLT2 inhibitors may promote glycosuria, increase the risk of UTI, and create favorable conditions for bacterial growth, some studies have shown that the risk of developing urosepsis was similar to that posed by the use of other anti-diabetics. A possible explanation for this is based on increased urinary flow due to osmotic diuresis and natriuresis [4,5,6].

Fournier’s gangrene (FG), a reported adverse event which may rarely occur with the use of SGLT2 inhibitors, is usually a fulminating, rare, life-threatening disease, with an occurrence estimated at 1.6/100,000 males per year and a mean age of appearance between 50 and 79 years. It is often a polymicrobial infection and may lead to the development of a fungal superinfection; therefore, broad-spectrum antibiotics to cover the most common microorganisms need to be administered and screening for fungal infections needs to be performed since the mortality rate in FG cases is high, reaching 67%. Early intervention is thus crucial [7]. A glycosuria-mediated bacterial-growth-inducing environment may be the key precursor to the development of infection. The entrance gate can be an abrasion, an open cut, or poor genital hygiene. Bacteremia triggers proinflammatory cytokines with affect the endothelium and initiate a coagulation cascade, inhibiting fibrinolysis, which, via thromboplastin, leads to microthrombus formation [4,5,6]. To our knowledge, there have been no cases of FG reported in Romania since SLGT2 inhibitors were approved.

## 2. Case Presentation

### 2.1. History of Presentation

We report the case of a 51-year-old male who presented to the emergency department (ED) with urinary symptoms and swollen testicles and was admitted to the intensive care unit (ICU) in a septic state. Previous history revealed chronic coronary syndrome and atrial fibrillation with a high ventricular response rate, with onset at approximately one month prior to admission, simultaneously with the appearance of a respiratory intercurrence (acute pneumopathy) for which he received cefixime for 7 days at discharge. It should also be noted that the patient has a history of smoking and alcohol consumption.

After two weeks of antibiotics, he was hospitalized in the cardiology ward with congestive heart failure. Transthoracic echocardiography (TTE) showed left ventricular hypertrophy (LVH) with severely impaired systolic function [left ventricular ejection fraction (LVEF) = 10–15%], with severe global hypokinesia, akinesia in the basal 1/3 of interventricular septum (IVS), left atrial enlargement, restrictive LV filling patterns, normal LV mass, cardiac output (CO) = 2.7 L/min, cardiac index (CI) = 1.4 L/min/m^2^, stroke volume (SV) = 39 mL/beat, indexed SV = 20 mL/beat/m^2^, a moderate ischemic and degenerative mitral regurgitation, a moderate tricuspid regurgitation, mild secondary pulmonary hypertension, and impaired right ventricular (RV) longitudinal systolic function.

Coronary angiography was performed due to the patient’s coronary history and in order to identify a potential cause for the severe LV dysfunction. However, the lesions detected did not provide an explanation [indomitable left coronary trunk, diffusely atheromatous left anterior descending artery (LAD) with 40% proximal stenosis, circumflex artery (CxA) with 50–70% ostial stenosis and 50–70% distal stenosis, right coronary artery (RCA) with chronic proximal occlusion with distal loading up to the level of the stent through an important epicardial collateral from the first diagonal artery.

Given the intercurrence episode, the fact that coronarography did not explain the severe dysfunction, and taking into account the patient’s stationary evolution under optimal drug therapy, the suspicion of acute myocarditis was raised and corticosteroid pulse therapy using intravenous (iv.) methylprednisolone was administered, followed by an oral dose and introduction of levosimendan on the automatic syringe at a dose of 1.6 mL/h, 0.05 mcg/kg/min, for 24 h, while maintaining hemodynamic stability. Nevertheless, angiotensin receptor–neprilysin inhibitor (ARNi) and SGLT2 inhibitor therapy was initiated according to the newest guidelines and was well tolerated during hospitalization. The patient was discharged with the indication of performing a gadolinium MRI, which confirmed the diagnosis of myocarditis. A late gadolinium enhancement sequence revealed linear myocardial fibrosis in the medio-ventricular segment of the interventricular septum and the medio-basal segment of the anterior wall as well as transmural fibrosis in the apical segment of the inferior wall (Figure 1, Table 1). A few days after discharge from the cardiology ward, he complained about pain in the scrotal area, but he delayed seeking medical attention and presented into the ED after two weeks.

### 2.2. Investigations

The physical examination revealed painful scrotal edema, palpably distended bladder, with suprapubic tenderness. Blood analysis showed acute retention of urine, leukocytosis with neutrophilia, and elevated C reactive protein. Urinalysis showed glycosuria and leukocyturia. Additional laboratory parameters are presented in Table 2.

The calculated LRINEC (Laboratory Risk Indicator for Necrotizing Fasciitis) score was 8, with high risk of necrotizing fasciitis and a positive predictive value of 93.4%. After 3 days, the scrotal edema extended to the suprapubic area and the left inguinal region with subcutaneous crepitations up to the left subcostal region. A culture was harvested the from wound secretion and the results showed multi-drug-resistant Klebsiella pneumoniae ssp. pneumoniae. An abdominal and pelvic computed tomography (CT) scan was performed, which revealed subcutaneous emphysema extending from the left scrotal region to the left anterolateral side of the abdomen and the left lateral side of the chest up to the left costal margin, consistent with the diagnosis of FG.

### 2.3. Management (Medical/Interventions)

First of all, SGLT2 inhibitor intake was discontinued. The patient received intravenous antibiotics and fluids for sepsis management. Surgical exploratory intervention was performed and showed periprostatic purulent collection with extension to the perineum. At the level of the left inguinal canal, a wide incision was made on the muscle fascia. Lavage of the area with betadine and oxygenated water was performed and excision of necrotic tissues was conducted, stopping the evolution of necrotizing fasciitis Figure 2.

## 3. Discussion

This case of FG is the first one that we have experienced so far in association with novel glucose-lowering agents. According to various studies reported, time to onset of FG after initiation of SGLT2 inhibitors ranged from 5 days to 49 months, which applies to our case as well (2 weeks) [8]. We performed a brief search of the literature published from 2016 to 2023 using the PubMed database using the following keywords: “Fournier’s gangrene”, “sodium glucose cotransporter 2 inhibitors”, and “serious adverse events”, and identified 16 case reports of Fournier’s gangrene in association with SGLT2 inhibitors, as summarized in Table 3.

The median age in these reports was 59, with a majority of patients being male (12 vs. 4). Nine cases included obese patients and six included current or ex-smokers. All case reports found, excluding the current one, included patients with T2DM. Dapagliflozin usage was found in 10 cases, empagliflozin in four cases, and canagliflozin in two cases. Onder et al. presented the case of a 64-year-old male with T2DM who developed FG after 6 months of dapagliflozin treatment [21]. Kumar et al. described a 41-year-old male with T2DM that developed FG after 7 months of empagliflozin administration [23]. Chi WC et al. reported a 67-year-old male with symptoms developed 3 weeks after dapagliflozin was initiated [24]. Ellegård et al. reported a 52-year-old female with T2DM taking dapagliflozin for 1.5 years who was prescribed prednisolone after a past adrenalectomy. The patient was obese and was a smoker [14]. As Jeff Yufeng Yang et al. stated in their research, real-world evidence on the incidence of FG associated with SGLT2 inhibitors is lacking. They analyzed 517.000 patients either taking SGLT2 inhibitors or non-SGLT2-inhibitor antihyperglycemic drugs using a state-of-the-art ACNU study design for real-world evidence, which showed no significant statistical crude FG incidence rates between cohorts (3.8 vs. 3.2) [25].

Clinical data provided by AstraZeneca Pharmaceuticals regarding the occurrence of FG in populations taking SGLT2 inhibitors are now available. Reviewing the FAERS database from 2013 to 2020, 491 cases of FG related to SGLT2 inhibitors were found, with the majority encountered with empagliflozin, canagliflozin, and dapagliflozin (223 vs. 162 vs. 101); the remaining 5 cases included ertugliflozin [26]. Going further back, from 2004 to 2018, 47 cases were found, suggesting that this relationship is more of an association rather than causality. Using the same database, we investigated evidence up to June 30, 2023, and identified 1.032 cases of FG in association with the use of SGLT2 inhibitors (dapagliflozin = 121, empagliflozin = 672, canagliflozin = 246 and ertugliflozin = 5 cases) as well as 270 cases of necrotizing fasciitis (dapagliflozin = 52, empagliflozin = 133, canagliflozin = 52), in 60 cases out of which the patients died [26].

The US Food and Drug Administration (FDA) reported twelve patients treated with SGLT2 inhibitors who, after months of treatment, developed a rare and serious genital infection called necrotizing fasciitis or Fournier’s gangrene, of whom one died [27]. Canagliflozin administration was associated with an increased risk of leg and foot amputations, as shown in the CANVAS trial and CANVAS-R trial; compared with placebo, patients treated with canagliflozin developed these types of complications twice as often and had an increased risk of bone fractures and decreased bone mineral density [28]. Over a period of three years, 101 patients developed acute kidney injury after receiving canagliflozin [29]. Further trials demonstrated major cardiovascular and renal benefits and even though the risk of amputation remains, it has been proven that it is lower than previously reported and can be reduced with adequate monitoring [28]. Ketoacidosis was reported in 73 cases; 19 cases of life-threatening blood infections (urosepsis) and kidney infections (pyelonephritis) that started as urinary tract infections were identified [30].

The European Medicines Agency’s Pharmacovigilance Risk Assessment Committee (EMA/PRAC) also investigated the relationship between these two variables and raised awareness of the possible causality, therefore requiring proper labeling information according to the newest data available.

The approval of these novel glucose-lowering agents began in 2013 and since then, six SGLT2 inhibitors have been approved for the treatment of T2DM by the FDA and EMA: empagliflozin, dapagliflozin, canagliflozin, ertugliflozin, bexagliflozin, and sotagliflozin [31]. Their cardioprotective mechanisms can be explained by an increase in distal tubular sodium load, inhibiting the renin–angiotensin system (RAS) and reducing preload (natriuresis and diuresis) and afterload (arterial vasodilatation). Moreover, their pleiotropic effects target the endothelium, reducing oxidative stress, lowering uric acid values, and improving arterial stiffness. SGLT2 inhibitors promote better heart efficiency and function based on the “thrifty substrate” hypothesis, according to which ketones become the cardiac fuel source. Ketogenesis is favored by the increase in glucagon production, and beta-hydroxybutyrate, which has antiarrhythmic effects, becomes the main source of energy [4,32].

Their nephroprotective mechanisms are also based on distal sodium load. By inhibiting tubuloglomerular feedback, intraglomerular pressure will drop due to vasoconstriction in the afferent arteriole, which will result in decreased albuminuria. Natriuresis, besides its effects on preload, will also reduce the effective circulating volume, blood pressure, and weight [32].

Predisposing factors for the development of Fournier’s gangrene include alcoholism, tobacco use, systemic disorders, obesity, hypertension, diabetes mellitus (20–70%), chronic renal insufficiency, congestive heart failure, malignant neoplasms, human immunodeficiency virus infection, and corticosteroid intake, along with poor genital hygiene [5]. In contrast to these data, our patient was of a young age, was neither hypertensive nor obese, and did not have a history of T2DM, but did have a history of smoking and alcohol abuse. According to data found in the literature, 25–50% of FG cases are associated with alcohol intake, and the rest with immunosuppression [33,34]. The pathophysiological process underlying the progression of the disease is based on hypoxia and low blood supply, which create a favorable environment for anaerobic microorganisms and also make it harder for antibiotics to reach their site of action. More specifically, endarteritis obliterans occurs and small subcutaneous vessels suffer thrombosis. Susceptibility is even higher if the patient has associated comorbidities which alter the cell-mediated immune response and vascular system. The Fournier Gangrene Severity Index (FGSI) was developed in order to determine the severity and prognosis of the disease. A score >9 is associated with a 75% probability of death. In our case, the FGSI score calculated using available laboratory parameters (serum creatinine, hematocrit, white blood cell (WBC) count) was at least 9 [34].

Supporting our findings is a case of a non-diabetic patient with CKD who developed FG 23 months after the initiation of SGLT2 inhibitors. Surgical debridement was performed, and an intraoperative swab showed *Staphylococcus lugdunensis* [35]. For patients without diabetes, the available data are limited, as most trials that used SGLT2 inhibitors included patients with T2DM. Out of all HF and CKD trials, data regarding patients without DM were available from two HFrEF trials (EMPEROR REDUCED, DAPA-HF), one HFpEF trial (EMPEROR-PRESERVED), and one CKD trial (DAPA-CKD) [36]. Mycotic genital infections were reported in 371 cases, and UTIs were reported in 2471 cases. The risk of UTIs was 7% higher and the risk of mycotic genital infections was 3.54 times higher in the SGLT2 group, but these adverse events rarely resulted in serious complications and there were insufficient cases of FG to accurately determine the relative risk (RR). The DAPA-HF trial reported a single case of FG in the placebo arm, and in the DECLARE TIMI-58 trial, five cases were reported in the placebo arm compared to only one in the dapagliflozin arm. The DAPA-HF trial reported a significantly lower number of UTIs (11 dapagliflozin vs. 17 placebo) compared to others. The EMPEROR-Reduced and EMPEROR-Preserved trials reported UTIs in 73 and 261 non-diabetic patients (of whom 188 received empagliflozin vs. 146 placebo). Genital infections were also found in 21 and 38 non-diabetic patients (43 received empagliflozin vs. 16 placebo) [36,37].

## 4. Conclusions

Although to a lesser extent than patients with diabetes, non-diabetic patients also present a risk of developing infections, and taking into consideration that our subject had a history of smoking and alcohol consumption, the association of predisposing factors may have contributed to an increased risk of developing FG. The distinguishing feature of this case is that the patient was not diabetic, which emphasizes that patients without diabetes who are treated for heart failure with SGLT2 inhibitors may also develop urogenital infections. The patient’s prognosis improved through early treatment and collaboration between urologists and cardiologists. Serious adverse events need to be reported voluntarily in order to verify the relationship and minimize the risk of bias. Serial assessments and evaluations such as screening for urinary infections must be performed, and patients must be educated to help them recognize the symptoms and achieve optimal genital hygiene in order to reduce the risk of infections. The incidence of SGLT2 inhibitor-associated FG is almost impossible to report and, therefore, several limitations still play an important role in establishing whether the relationship is one of causality than an association. In most cases, it is impossible to accurately measure the frequency of these reactions or demonstrate a causal association with drug exposure since these events are self-reported, occur in a population of unknown size, and are not statistically significant. However, the benefits of prescribing SGLT2 inhibitors outweigh the risks, and since these agents represent a novel therapy with proven cardiovascular and renal benefits, it is important to give the best patient-centered interventions in order to receive optimal outcomes.

## Figures and Tables

**Figure 1 medicina-60-00837-f001:**
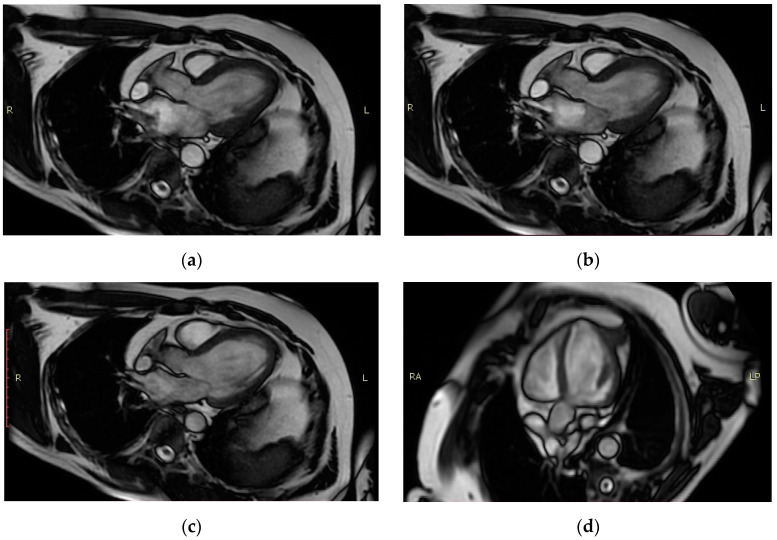
**Magnetic resonance imaging** (**a**–**d**) showed global hypokinesia predominantly of the interventricular septum, the apical segment of the inferior wall, and the medio-apical segment of the anterior wall, consistent with inflammatory dilatative cardiomyopathy, post-myocarditis, and post-infarction sequelae.

**Figure 2 medicina-60-00837-f002:**
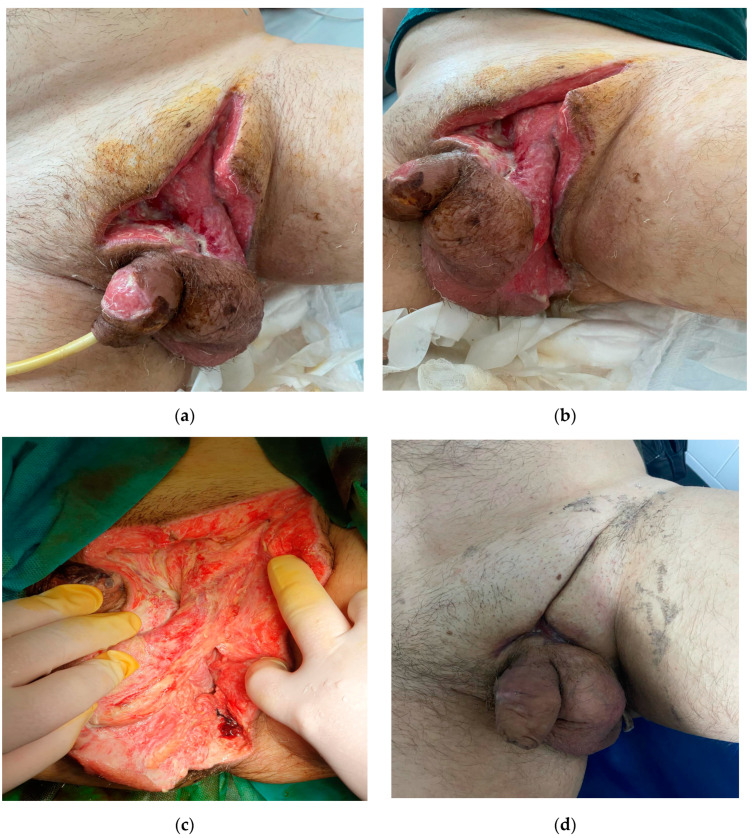
**Surgical evolution** (**a**–**d**). Surgical approach and the healing process during hospitalization.

**Table 1 medicina-60-00837-t001:** Magnetic resonance imaging parameters.

Left Ventricle (LV)		Right Ventricle (RV)	
LV ejection fraction	30%	RV ejection fraction	48%
Stroke volume	48.2 mL	Stroke volume	51.3 mL
End-diastolic (ED) volume	160 mL	End-diastolic volume	106 mL
End-systolic (ES) volume	112 mL	End-systolic volume	54.6 mL
Heart rate (HR)	48 bpm	HR	48 bpm
Mass ED	83 g	Standard deviation HR	1 bpm
Cardiac output	2.3 L/min		
Mass	83 g		
Mass ES	88 g		
End-diastolic epicardial volume	239 mL		
End-systolic epicardial volume	196 mL		
Standard deviation HR	1 b pm	Flow Qp/Qs	0.967
Ant. sep. wall thickness	0.9 cm		
Inf. lat. wall thickness	0.7 cm		
End-diastolic dimension	5.8 cm		
End-systolic dimension	4.8 cm		
Fractional shortening	16%		

**Table 2 medicina-60-00837-t002:** Laboratory parameters.

Blood Tests—Venous Blood Testing		Value
**Blood cell count**	**Leukocytes**	**40.200/µL**
	**Neutrophiles**	**87.900/µL**
	Hemoglobin	17.6 g/dL
**Chemistry**	**Serum Creatinine**	**3.29 mg/dL**
	**Urea**	**165.93 mg/dL**
	Serum Glucose	104.58 mg/dL
**Serology**	Procalcitonin	0.49 µg/L
	**C reactive protein**	**57.78 mg/L**
**Urinalysis** **—Urine sediment examination**		**Value**
	**Leukocytes**	**25/µL**
	**Glucose**	**50 mg/dL**
	**Ascorbic acid**	**20 mg/dL**
	Proteins	-
	Ketones	-
	Biliary pigments	-
	Red blood cells	-

* Abnormal values are bolded.

**Table 3 medicina-60-00837-t003:** Brief summary of case reports of Fournier’s gangrene associated with SGLT-2 inhibitors [2,3,4,5,6,7,8,9,10,11,12,13,14,15,16,17].

Year	Article	Gender	Age	SGLT2 Inhibitor	Onset	Obesity	Smoking	T2DM
2023	This paper	Male	55	Dapagliflozin	2 weeks	No	No	No
2022	Jahir et al. [9]	Female	58	Empagliflozin	4 months	Yes	-	Yes
2021	Elbeddini et al. [10]	Female	71	Dapagliflozin	5 years	Yes	-	Yes
2021	Vargo et al. [11]	Male	64	Dapagliflozin	-	-	-	Yes
2021	Moon et al. [12]	Male	66	Dapagliflozin	3 years	-	Yes	Yes
2021	Newton et al. [13]	Male	41	Dapagliflozin	NR	-	-	Yes
2020	Ellegård et al. [14]	Female	52	Dapagliflozin	1.5 years	Yes	Yes	Yes
2020	Kasbawala et al. [15]	Female	37	Canagliflozin	1 month	Yes	-	Yes
2020	García et al. [16]	Male	68	Dapagliflozin		-	-	Yes
2020	Elbeddini et al. [17]	Male	72	Canagliflozin	6 years	-	Yes	Yes
2019	Nagano et al. [18]	Male	34	Empagliflozin	5 months	-	-	Yes
2019	Rodler et al. [19]	Male	39	Dapagliflozin	4 years	Yes	Yes	Yes
2019	Elshimy et al. [20]	Male	57	Empagliflozin	10 days	Yes	-	Yes
2019	Onder et al. [21]	Male	64	Dapagliflozin	6 months	Yes	Ex-smoker	Yes
2018	Omer et al. [22]	Male	60	Dapagliflozin	4 months		-	Yes
2017	Kumar et al. [23]	Male	41	Empagliflozin	7 months	Yes	Yes	Yes
2016	Chi et al. [24]	Male	67	Dapagliflozin	3 weeks	Yes	-	-

NR = not reported.

## Data Availability

Data presented in this study are available upon request from the corresponding author due to patient confidentiality.

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
