# Peer review of "Fournier’s Gangrene as an Adverse Event Following Treatment with Sodium Glucose Cotransporter 2 Inhibitors"

_medicina, 2024, doi:10.3390/medicina60050837_

Round 1

Reviewer 1 Report

Comments and Suggestions for Authors

This manuscript needs to be re-written in a more concise and focused way, it should be shortened and deal only with the occurrence of Fournier’s gangrene as an adverse event in a patient with heart problems under sodium-glucose cotransporter-2 inhibitors treatment. These comments apply to all sections of the manuscript (Introduction, Case presentation, Discussion and Conclusions).

Other specific comments include:

1) Give abbreviations in full when first mentioned (SGLT2, T2DM, FG in the Abstract and text, and ER, TTE and many others in the presentation of the case and in the discussion).

Please be consistent with the abbreviations (either SGLT2 or SGLT-2)

2) The introduction does not seem very relevant (mechanisms of renal expression glucose) to the case reported and should be re-written.

3) Paragraph 2.1. (History of presentation) should be more concise and clear.

4) Paragraph 2.3. (Differential diagnosis) does not belong to the case presentation.

5) Table 2. Laboratory parameters (where in the urine? In the serum?).

6) The conclusions should be shortened by avoiding parts of the discussion or of the case report and should be based strictly on the findings of the manuscript..

7) Linguistic editing is necessary (see old inferior myocardial infarction, There is also to be mentioned that the patient is noncompliant to medical, A native CT scan, emphysema bubbles that migrate, costal arch, abdominal and thoracic soft parts, of case reports developing Fournier’s gangrene (Table 3), as summarized in table 2 (Discussion) should be Table 3.

Comments on the Quality of English Language

Linguistic editing is necessary (see old inferior myocardial infarction, There is also to be mentioned that the patient is noncompliant to medical, A native CT scan, emphysema bubbles that migrate, costal arch, abdominal and thoracic soft parts, of case reports developing Fournier’s gangrene (Table 3) and so on.

Author Response

Thank you very much for taking the time to review this manuscript. Please find the detailed responses below and the corresponding revisions and corrections in the re-submitted file.

Comment 1: [This manuscript needs to be re-written in a more concise and focused way, it should be shortened and deal only with the occurrence of Fournier’s gangrene as an adverse event in a patient with heart problems under sodium-glucose cotransporter-2 inhibitors treatment. These comments apply to all sections of the manuscript (Introduction, Case presentation, Discussion and Conclusions).

Response 1: We agree with this comment, therefore, we have revised the entire manuscript: the abstract (which is now based only on the case), the introduction is shorter (we deleted the pathophysiology and we focused only on the genitourinary adverse events). The paragraph “history of presentation” has been rewritten (previous manuscript included “2.2 past medical history” and in order to have a clearer history we have encapsulated it in “2.1 history of presentation). “4. Conclusions” have been rewritten and based only on the findings of the manuscript.

Comment 2: [Give abbreviations in full when first mentioned (SGLT2, T2DM, FG in the Abstract and text, and ER, TTE and many others in the presentation of the case and in the discussion). Please be consistent with the abbreviations (either SGLT2 or SGLT-2)]

Response 2: Agree. We have, accordingly, modified the abbreviations.

Comment 3: [Paragraph 2.1. (History of presentation) should be more concise and clear]

Response 3: The paragraph “history of presentation” has been rewritten (previous manuscript included “2.2 past medical history” and in order to have a clearer history we have encapsulated it in “2.1 history of presentation).

Comment 4: [Paragraph 2.3. (Differential diagnosis) does not belong to the case presentation]

Response 4: We have deleted the paragraph from the case presentation and moved it in the introduction where we focused only on the genitourinary adverse events.

Comment 5: [Table 2. Laboratory parameters (where in the urine? In the serum?)]

Response 5: We hope we understood the asignment. If this is not appropriate, please let us know.

Comment 6: [The conclusions should be shortened by avoiding parts of the discussion or of the case report and should be based strictly on the findings of the manuscript.]

Response 6: We have indeed shortened the conclusions and erased the parts that were found in the other sections.

Comment 7: [Linguistic editing is necessary (see old inferior myocardial infarction, There is also to be mentioned that the patient is noncompliant to medical, A native CT scan, emphysema bubbles that migrate, costal arch, abdominal and thoracic soft parts, of case reports developing Fournier’s gangrene (Table 3), as summarized in table 2 (Discussion) should be Table 3]

Response 7: old inferior myocardial infarction -> we only left “chronic coronary syndrome”

There is also to be mentioned that the patient is noncompliant to medical -> It should also be noted that the patient has a history of smoking and alcohol consumption.

A native CT scan -> An abdominal and pelvic computed tomography (CT) scan

emphysema bubbles that migrate, costal arch, abdominal and thoracic soft parts,-> subcutaneous emphysema extending from the left scrotal region to the left anterolateral side of the abdomen and the left lateral side of the chest up to the to the left costal margin.

of case reports developing Fournier’s gangrene (Table 3) -> Brief summary of case reports of Fournier’s gangrene associated with SGLT-2 inhibitors

as summarized in table 2 (Discussion) should be Table 3-> we modified the error

Point 1: Linguistic editing is necessary (see old inferior myocardial infarction, There is also to be mentioned that the patient is noncompliant to medical, A native CT scan, emphysema bubbles that migrate, costal arch, abdominal and thoracic soft parts, of case reports developing Fournier’s gangrene (Table 3) and so on.

Response 1: After correcting all the potential errors, we hope that this version of the manuscript will no longer require editing.

Reviewer 2 Report

Comments and Suggestions for Authors

Fournier's gangrene is a well established adverse event which may rarely occur with use of SGLT2 inhibitors due to associated glucosuria predisposing to bacterial infection.  This is a case report with a brief literature review.   What is new and merits reporting is that this man did not have premorbid diabetes, apparently.   Clearly, the authors need to try to encapsulate this as a short case report to show that patients without diabetes, treated for renal failure or congestive heart failure, may also develop Fournier's gangrene.  A major rewrite is in order.

SPECIFIC COMMENTS  please avoid the melodrama of "Deafening Silence"

Comments on the Quality of English Language

English is fine

Author Response

Thank you very much for taking the time to review this manuscript. Please find the detailed responses below and the corresponding revisions in the re-submitted file.

Comment 1: [Fournier's gangrene is a well established adverse event which may rarely occur with use of SGLT2 inhibitors due to associated glucosuria predisposing to bacterial infection. This is a case report with a brief literature review.  What is new and merits reporting is that this man did not have premorbid diabetes, apparently. Clearly, the authors need to try to encapsulate this as a short case report to show that patients without diabetes, treated for renal failure or congestive heart failure, may also develop Fournier's gangrene. A major rewrite is in order.]

Response 1: Thank you for reviewing our article. We agree with this comment and therefore we have rewritten the manuscript including the abstract (which is now based only on the case), the introduction is shorter (we deleted the pathophysiology and we focused only on the genitourinary adverse events). The paragraph “history of presentation” has been rewritten (previous manuscript included “2.2 past medical history” and in order to have a clearer history we have encapsulated it in “2.1 history of presentation). “4. Conclusions” have been rewritten and based only on the findings of the manuscript.

Comment 2: [please avoid the melodrama of "Deafening Silence"]

Response 2: We have, accordingly, modified the title.

Round 2

Reviewer 1 Report

Comments and Suggestions for Authors

No further comments

Author Response

Thank you so much for your time and for making this a better article. 

Reviewer 2 Report

Comments and Suggestions for Authors

The authors have improved their presentation.    They have only at conclusion pointed out the unique feature of their case.  This was a person without diabetes who developed Fournier's gangrene.   So the title is misleading in indicating ":glucose lowering agents" when it actually should be SGLT2 inhibitors.  Their literature search must focus on Fourniers gangrene in patients without diabetes prescribed SGLT2 inhibitors for heart failure or renal failure.

Author Response

Thank you so much for taking the time to review this manuscript. Please find the detailed responses below and the corresponding corrections in the re-submitted file.

Comment 1: the title is misleading in indicating ":glucose lowering agents" when it actually should be SGLT2 inhibitors

Response 1: The new title is: Fournier’s gangrene as an adverse event following sodium glucose cotransporter 2 inhibitors

Comment 2: Their literature search must focus on Fourniers gangrene in patients without diabetes prescribed SGLT2 inhibitors for heart failure or renal failure.

Response 2: We searched the online database and found only one case of FG in a patient with CKD without diabetes and therefore we checked large RCT in order to find genitourinary adverse events reported there. A new paragraph has been added to the discussion chapter.
